# Diverse Genetic Landscape of Suspected Retinitis Pigmentosa in a Large Korean Cohort

**DOI:** 10.3390/genes12050675

**Published:** 2021-04-30

**Authors:** Yoon-Jeon Kim, You-Na Kim, Young-Hee Yoon, Eul-Ju Seo, Go-Hun Seo, Changwon Keum, Beom-Hee Lee, Joo-Yong Lee

**Affiliations:** 1Department of Ophthalmology, Asan Medical Center, University of Ulsan College of Medicine, Seoul 05505, Korea; anne215@gmail.com (Y.-J.K.); youna.kim.oph@gmail.com (Y.-N.K.); yhyoon@amc.seoul.kr (Y.-H.Y.); 2Department of Laboratory Medicine, Asan Medical Center, University of Ulsan College of Medicine, Seoul 05505, Korea; ejseo@amc.seoul.kr; 33billion Inc., Seoul 05505, Korea; ghseo@3billion.io (G.-H.S.); ckeum@3billion.io (C.K.); 4Medical Genetics Center, Asan Medical Center Children’s Hospital, University of Ulsan College of Medicine, Seoul 05505, Korea

**Keywords:** retinitis pigmentosa, inherited retinal diseases, whole exome sequencing, targeted next-generation sequencing

## Abstract

We conducted targeted next-generation sequencing (TGS) and/or whole exome sequencing (WES) to assess the genetic profiles of clinically suspected retinitis pigmentosa (RP) in the Korean population. A cohort of 279 unrelated Korean patients with clinically diagnosed RP and available family members underwent molecular analyses using TGS consisting of 88 RP-causing genes and/or WES with clinical variant interpretation. The combined genetic tests (TGS and/or WES) found a mutation in the 44 RP-causing genes and seven inherited retinal disease (IRD)-causing genes, and the total mutation detection rate was 57%. The mutation detection rate was higher in patients who experienced visual deterioration at a younger age (75.4%, age of symptom onset under 10 years) and who had a family history of RP (70.7%). The most common causative genes were *EYS* (8.2%), *USH2A* (6.8%), and *PDE6B* (4.7%), but mutations were dispersed among the 51 RP/IRD genes generally. Meanwhile, the *PDE6B* mutation was the most common in patients experiencing initial symptoms in their first decade, *EYS* in their second to third decades, and *USH2A* in their fifth decades and older. Of note, WES revealed some unexpected genotypes: *ABCC6*, *CHM*, *CYP4V2*, *RS1*, *TGFB**I*, *VPS13B*, and *WDR19*, which were verified by ophthalmological re-phenotyping.

## 1. Introduction

Inherited retinal diseases (IRDs) are a group of heterogenous conditions in which progressive visual impairment is caused by retinal degeneration and are mainly caused by Mendelian mutations in 1 out of at least 300 genes, and the prevalence of IRDs is expected as about 1:1000 in the East Asian populations [1,2]. Retinitis pigmentosa (RP) is the most common IRD which over 1 million patients worldwide are affected—approximately 1 in 4000 individuals—and leads to legal blindness in the advanced stage [3]. More than 80 genes have been identified as being responsible for RP [1,4]. Diverse functions of RP causative genes involve various pathways, i.e., phototransduction, vitamin A metabolism, signaling, cell–cell interaction, and protein synthesis, i.e., structural or cytoskeletal proteins, synaptic interaction proteins, mRNA intron-splicing factors, trafficking of intracellular proteins, maintenance of cilia/ciliated cells, phagocytosis, pH regulator and a few encode proteins with yet-unknown function [5]. In addition, genetic abnormalities expressed in various organs other than the eyes cause syndromic RP such as Usher syndrome [6]. In relation to these diverse functions of causative genes, considerable ethnic and regional differences are expected in its genetic landscape [7,8]. 

Genetic diagnosis is important for adequate genetic counseling and visual prognosis prediction in patients with RP. However, genetic diagnosis of RP is challenging and time consuming due to its genetic heterogeneity. Recent advances in genetic testing have enabled the identification of disease-causing genes in a cost-effective and time-efficient way with sufficient accuracy. Massive parallel sequencing techniques such as targeted next-generation sequencing (TGS) and whole exome sequencing (WES) offer a better opportunity for genetic analysis. In the current study, we used TGS and/or WES to assess molecular genetic profiles in the largest-to-date number of Korean patients with RP to determine their genetic distributions. TGS yielded a 45% mutation detection rate, but WES enhanced the detection rate by 10% and identified IRDs other than RP, expanding the genetic spectrum of RP and its related disorders in the Korean patients.

## 2. Materials and Methods

### 2.1. Patients

We conducted this observational open-label cohort study at a single tertiary clinic, the Asan Medical Center (Seoul, Korea). The study was conducted according to the tenets of the Declaration of Helsinki, and all study-related data acquisitions were approved by the Institutional Review Board of Asan Medical Center (AMC IRB No. 2019-0106). Informed consent was obtained from each patient prior to enrollment. Between March 2018 and July 2020, a total of 279 unrelated Korean probands, clinically diagnosed with RP, underwent molecular diagnosis as described in Figure 1. 

### 2.2. Clinical Diagnosis of Retinitis Pigmentosa 

All patients underwent detailed dilated-pupil ophthalmologic examinations including measurement of best-corrected visual acuity (BCVA), slit-lamp biomicroscopy, funduscopic examination, fundus photography, fundus autofluorescence imaging (Optos, Dunfermline, UK), and full-field electroretinography (Roland-Consult, Brandenburg, Germany) to confirm the diagnosis of RP, according to the standards of the International Society for Clinical Electrophysiology of Vision. We examined spectral domain optical coherence tomography (OCT) (Heidelberg, Dossenheim, Germany) images to evaluate the retinal structures. Two retinal specialists (Y.J.K., Y.N.K.) confirmed all clinical data and the accuracy of diagnosis. In all the recruited patients with RP who received a genetic test, we phenotypically reclassified them on the basis of genetic findings and repeated clinical assessment according to their genotypes. 

### 2.3. Analysis of Genetic Variants 

We extracted genomic DNA from peripheral blood samples taken from the patients. We performed TGS for 220 patients from 170 families using the Ion Torrent S5XL platform (Thermo Fisher Scientific Inc., Waltham, MA, USA) using a panel of 88 genes associated with RP (Appendix A). The mean depth of coverage was approximately 500-fold, with 99.2% coverage higher than 20-fold. We waived the verification of identified variants for Torrent S5XL sequencing data when the read depth was over 100 reads and the allele frequency was 40 to 60% [9]. Performing WES, we captured all exons of all genes (approximately 22,000) using a SureSelect kit (Version C2; Agilent Technologies Inc., Santa Clara, CA, USA). We sequenced the captured genomic regions using a NovaSeq platform (Illumina Inc., San Diego, CA, USA). Analysis of raw genome sequencing data included alignment to the reference sequence (NCBI genome assembly GRCh37; accessed in February 2009), and we performed variant calling, annotation, and prioritization as previously described [10]. We achieved validation of WES data with subsequent Sanger sequencing. 

We classified all identified variants according to the guidelines of the American College of Medical Genetics and Genomics (ACMG) [11]. We re-evaluated the pathogenicity of variants identified as variant of unknown significance (VUS) by familial segregation analysis. We manually reviewed the VUSs in patients who did not undergo segregation analysis based on their phenotypes consistent with RP and the correlation of posterior probabilities obtained using a Bayesian calculator [12].

### 2.4. Statistical Analysis 

We used descriptive statistics (number and percentage for categorical variables; mean and standard deviation for continuous variables) to summarize the baseline patient characteristics. For comparisons between patients with or without a family history of RP, we used the independent *t*-test, and *p*-values < 0.05 were considered statistically significant. We performed statistical analyses for these characteristics using SPSS Statistics for Windows, version 21 (IBM Corp., Armonk, NY, USA).

## 3. Results

### 3.1. Demographics

We recruited a total of 279 unrelated probands with RP; 170 probands underwent TGS and 109 probands underwent WES. Of 94 probands in whom we found no appropriate causative genes by TGS, we re-analyzed 75 by WES. The flowchart in Figure 1 demonstrates the recruitment progress for genetic testing in patients with RP. 

Table 1 presents the baseline characteristics of the 279 probands at the time of genetic testing. Among the 279 probands, 131 were male (47.0%) and 148 were female (53.0%). The mean age at genetic testing was 47.6 ± 15.7 years, and patients experienced the initial symptoms at the mean age of 25.6 ± 16.9 years. Ninety-two probands (33.0%) had a family history of RP. The initial symptoms occurred most often within the age range of 11 to 20 years (*N* = 82, 29.4%), followed by the groups aged less than 10 years (*N* = 69, 24.7%) and 21 to 40 years (*N* = 61, 21.9%). 

### 3.2. Genetic Distribution of RP in Korean Patients

Figure 2a describes the overall genetic spectrum and their detection rates are demonstrated in Table 2. In 145 of the 279 probands (54.1%), we found a total of 161 genetic variations in 39 RP-causing genes, and five genes causing IRDs other than RP, using either TGS or WES. According to the ACMG classification [11], we predicted 45 variants (28.0%) as pathogenic (PV), 55 variants (34.2%) as likely pathogenic (LPV), and 61 variants (37.9%) as VUSs (Appendix A). Among 181 detected variants, 43 variants were identified as novel variants (Appendix A). 

Familial segregation analysis was available in 60 families, and we confirmed 58 variants from the 60 families as causative: 18 variants (31.0%) predicted as PVs, 13 variants (22.4%) as LPVs, and 27 variants (46.6%) as VUSs. Among the 129 variants in 219 families without the segregation analysis, we predicted 31 variants (24.0%) as PVs, 48 variants (37.2%) as LPVs, and 50 variants (38.8%) as VUSs based on the clinical re-assessment of the association between a patient’s genotype and phenotype. In summary, we genetically diagnosed 161 of 279 families (57.7%) and found 44 RP-causing genes and seven IRD-causing genes (Appendix A): 47 variants (26.0%) of PVs, 58 variants (32.0%) of LPVs, and 76 variants (42.0%) of VUSs. 

In these genetic spectra, three genes were the most commonly observed: *EYS* (*N* = 23, 8.2), *USH2A* (*N* = 19, 6.8%), and *PDE6B* (*N* = 13, 4.7%).In detail, TGS identified a mutation in 21 causative genes in 76/170 families (44.1%) (Figure 2b); *EYS* (*N* = 17, 10.0%), *PDE6B* (*N* = 10, 5.9%), and *USH2A* (*N* = 10, 5.9%) were the most common. We re-examined 75 of 94 families with negative results by TGS (Figure 2c) using WES, which revealed genetic alterations in 28 probands (37.3%) in 17 RP-causing and five IRD-causing genes, and *USH2A* (*N* = 3, 4.0%) was found to be the most common causative gene. From the WES results for unsolved TGS cases, all the detected variants included in the TGS panel were also detected as VUSs from TGS, and we re-tested those corresponding cases to confirm their pathogenicity. In addition, WES revealed mutations in five IRD-causing genes, including *ABCC6*, *CHM*, *CYP4V2*, *RS1*, and *TGFBI*, including eight cases with inconclusive results. Among the 109 probands who underwent WES without TGS (Figure 2d), we found mutations in 25 genes in 59 probands (54.1%). *USH2A* (*N* = 7, 6.4%) and *EYS* (*N* = 6, 5.5%) were the most frequently found causative genes. Of note, we found mutations in four IRD-causing genes including *CHM*, *CYP4V2*, *VPS13B*, and *WDR19*, including two cases with inconclusive results.

### 3.3. Mutational Spectrum According to the Age at Initial Symptom

According to the age of symptom onset (Figure 3), the mutation detection rate as well as genetic spectra were different. In the group aged under 10 years (Figure 3a), the detection rate was 78.3%, and the most common gene was *PDE6B* (*N* = 8, 11.6%), followed by *CNGA1*, *EYS*, *PRPF31*, and *USH2A* (*N* = 3, 4.3% for all). In the group aged between 11 and 20 years (Figure 3b), the detection rate was 63.4% and the common genes were *EYS* (*N* = 10, 12.2%), *RP1* (*N* = 6, 7.3%), and *USH2A* (*N* = 5, 6.1%). In the group aged 21 to 40 years (Figure 3c), the detection rate was 50.8%, and the common genes were similar to those in the 11- to 20-year-old group: *EYS* (*N* = 6, 9.7%), *RP1* (*N*=4, 6.5%), and *USH2A* (*N* = 4, 6.5%). In the group aged over 41 years (Figure 3d), the detection rate was low at 49.2%, and the most common gene was *USH2A* (*N* = 7, 11.5%), followed by *EYS* (*N* = 4, 6.6%).

### 3.4. Mutational Spectrum According to Family History of RP

Clinical characteristics were compared according to the presence of the family history of RP (Appendix A). Ninety-one probands (32.6%) had a family history of RP, and they experienced visual symptoms at a younger age compared to those without family history (23.2 ± 15.5 vs. 26.8 ± 17.7, *p* = 0.011). In addition, the mutation detection rate was higher in the patients with a family history of RP (72.5% vs. 55.9%, *p* = 0.007; risk ratio, 2.087 [95% confidence interval, 1.213–3.591]). Though the age at genetic examination was not significantly different, patients with a family history of RP tended to have significantly worse BCVA. 

### 3.5. Patients with Unexpected Genotypes: Genes Causing IRDs Other than RP

In reviewing the results of genetic analysis by WES, there were some cases with unexpected phenotypes that did not correlate with RP for the *ABCC6*, *CHM*, *CYP4V2*, *RS1*, *TGFBI*, *VPS13B*, and *WDR19* genes. We re-assessed the clinical phenotypes by reviewing medical records as well as ophthalmic characteristics, as described below (Table 3 and Appendix A). 

A 24-year-old woman had pseudoxanthoma elasticum (PXE) with retinal pigmentary dystrophy due to maternal-originated *ABCC6*, and the patient has had no other identified systemic disorders to date (Appendix A).In an 18-year-old man with reticular pigmentary dispersions with chorioretinal atrophy, choroideremia caused by a *CHM* mutation (Appendix A) was diagnosed; we clearly confirmed maternal transmission from the segregation test. His mother showed a mild form of chorioretinal atrophy from an ophthalmologic study.In the case of a 52-year-old woman showing diffuse chorioretinal degeneration, we found an associated Bietti crystalline dystrophy (BCD) caused by *CYP4V2* (Appendix A). We retrospectively reviewed seven years of medical records and found yellow-white crystals on fundus photographs that correlated with BCD before severe retinal degeneration progressed.A 45-year-old man with diffuse atrophy in the right eye and sectoral retinal degeneration in the left eye had the *RS1* mutation (Appendix A). As he was blind from childhood due to chronic retinal detachment in the right eye according to medical records, fundus findings in the right eye showed diffuse atrophic changes indistinguishable from RP.We enrolled siblings aged 15 and 17 years who had one-year-old brothers with corneal dystrophy and concomitant retinal pigmentary degeneration and found a *TGFBI* mutation (Appendix A). In their pedigree analysis, many maternal relatives had poor vision with corneal dystrophies in common.In a 41-year-old woman showing macula-dominant diffuse retinal dystrophy with mental retardation, Cohen syndrome caused by *VPS13B* (Appendix A) was diagnosed.In a 22-year-old woman with retinal pigmentary changes, *WDR19*-related Senior-Løken syndrome (Appendix A) was diagnosed.

### 3.6. Patients with Inconclusive Results

Besides the unexpected IRD-related results, there were seven inconclusive cases with variants in six genes: *CAPN5*, *CEP290*, *CRB1*, *GNAT1*, *IMPG1*, and *SNRNP200* (Appendix A). These cases consisted of patients with the causative genes not compatible with the ocular presentations (eight cases) or when the results of segregation analysis revealed inappropriate inheritance patterns or incomplete penetrance (two cases).

## 4. Discussion

In accordance with the initial clinical diagnosis of RP, we performed genetic testing for 279 probands. We performed the genetic confirmation on the basis of the ACMG criteria, the familial segregation analysis, and clinical relevance. As a result, the overall diagnostic yield was 57.7% (43.5% and 54.1% using TGS and WES, respectively) and among 75 negative TGS results, 28 (37.3%) were positive on WES. Of the 44 RP-causing genes detected in this analysis from 161 probands, the *EYS* (8.2%), *USH2A* (6.8%), and *PDE6B* (4.6%) were the most frequent causative genes, with different frequency according to the age of symptom onset and presence of family history. However, in general, mutations were dispersed throughout the causative genes, without a noticeable predominance of certain genotypes. In addition, the results of genetic analysis revealed unexpected genotypes, which were re-verified in the clinical diagnoses by reviewing medical records. 

### 4.1. Genetic Distribution in Korean Patients with RP

The overall diagnostic yield in our study was similar to those of previous studies [13,14,15], but the composition ratios of frequently found causative genes were slightly different: *EYS* (8.2%) and *USH2A* (6.8%). From previous studies, *EYS* variants account for the largest portion of causative genes in the East Asian region, especially in Korea and Japan; *EYS* variants account for 20 to 30% of RP cases, and *USH2A* variants cause less than 10% [16,17,18]. On the other hand, *USH2A* variants make up 20 to 40% and *EYS* variants account for less than 10% of RP cases, among the total RP population in Western or European ethnicities [8,19,20] as well as Chinese or Taiwanese populations [7,21]. The ratios of *EYS* and *USH2A* in our study were slightly lower than those in previous studies, which shows the unique genetic characteristics of Korean patients even in comparison with those of the other East Asian nations [18]. 

### 4.2. Mutational Spectrum According to the Age at First Symptom

In the present study, patients experienced their first ocular symptom in their mid 20s, and RP was diagnosed at 41 years of age, which were generally consistent with those reported in a Korean nationwide population-based study [22] and previous studies in other ethnicities [23,24]. In addition, genetic distributions were varied according to the age of symptom onset; *PDE6B* variants were prominent in the group aged under 10 years, *EYS* variants in the group aged 11 to 20 years, *EYS* and *USH2A* variants in the group aged 21 to 40 years, and *USH2A* variants were frequent in the group aged over 40 years. These trends are in line with distributions for the age of ocular symptom onset according to each genetic variant [7,21,25]: most patients with *PDE6B* variants experienced the initial symptom before age 10, while patients with *EYS* and *USH2A* variants experienced the first ocular symptom in late adolescence or adulthood. 

Interestingly, we found that the ratio of genetically undiagnosed cases increased with increasing age of symptom onset. Considering that fundus findings evolve with aging and, therefore, can be shown as different phenotypes according to the patient’s age, clinicians must keep in mind this possible interference of aging in making a proper diagnosis. In particular, elderly patients are likely to show progressive atrophic changes triggered by non-genetic factors, e.g., infectious retinal vasculitis [26], drug-induced retinal toxicity [27], or autoimmune retinopathies [28]. Additionally, it is likely that causative genes can be identified through repeated testing along with continued development of the human genome database in the future. 

### 4.3. Mutational Spectrum According to Family History

The detection rate of confirmed genetic diagnosis in patients with family history (70.7%) was higher than that of sporadic cases (51.3%). This is consistent with previous study results showing a detection rate lower in sporadic cases than in familial cases. One reason is that many genes responsible for sporadic RP are yet unidentified [29], and another possibility is that the TGS panel does not include the gene responsible for sporadic RP. In addition, non-genetic factors may affect the sporadic cases [30].

### 4.4. Genetic Tests and Correction of Clinical Diagnosis from Unexpected Causative Genes

From this study, genetic testing revealed IRD-causing genes (*ABCC6*, *CHM*, *CYP4V2*, *RS1*, *TGFB**I*, *VPS13B*, and *WDR19*) in 15 patients, for whom the initial clinical diagnoses were refined. Most of those causative genes, except *TGFB**I*, were known to induce pigmentary retinopathy, which requires differential diagnosis from RP. In reviewing patients’ medical records including previous examinations, we found clinical characteristics for the differential diagnosis of RP. Along with the overlapping clinical phenotypes of retinal degeneration, the common clinical presentations in the advanced stages of RP and IRDs hindered a clear clinical decision according to the initial phenotypes. Taken together, these results emphasize that patients’ detailed histories, family histories, and reassessments based on genetic examinations may help clinicians make IRD-compatible diagnoses precisely.

The ocular phenotypes of PXE caused by *ABCC6* mutations are variable but may show a ‘peau d’orange’ fundus appearance in childhood with reticular pigmentary dystrophy and crystalline bodies underlying the lesion of retinal pigment epithelium (RPE) atrophy [31]. Our case showed yellowish mottled features initially considered as RPE pigmentation by the clinician.In choroideremia, the retina covered with pigmentary changes evolves into areas of atrophy, especially in the mid-peripheral retina [32]. Our patients carrying *CHM* variants showed pigmented clumps with RPE degeneration as RP phenotypes; however, they also developed petalloid pattern atrophic plaques, the characteristic findings of choroideremia.The *CYP4V2* variants induce BCD characterized by multiple glistening intraretinal crystals scattered throughout the posterior poles of the eyes [33]. As noted in our case, since the crystals rarely become visible on the fundus examination in the advanced stages of retinal atrophy, en-face OCT images may be helpful for an accurate diagnosis in such cases.Retinoschisis is characterized by foveal retinal splitting and peripheral changes, with retinal pigmentations and vascular attenuation or sheathing, which can resemble RP [34]. Our patient demonstrated diffuse and sectoral RPE atrophy with a history of chronic retinal detachment, which was mis-interpreted as RP combined with cystoid macular edema.Cohen syndrome is an uncommon systemic disease caused by a *VPS13B* variant, presenting with mental impairment and retinal dystrophy [35]. Even though our patient had mental impairment, diagnosis of Cohen syndrome was possible after the confirmation of genetic analysis.Senior-Loken Syndrome, affecting the kidney and retina, leads to nephronophthisis, Calori disease, and RP [36]. Though a patient may show no clinical renal or hepatic diseases, a combination of clinical findings and genetic testing improves the accuracy of diagnoses in syndromic diseases.

Other than the six RP-associated genes described above, the siblings with *TGFB**I* mutation in this study showed fundus findings similar to those of RP. *TGFB**I*, a major causative gene for corneal dystrophies [37], was not clearly identified as associated with RP. Recently, several studies reported that *TGFB**I* variants and *TGFBR1* polymorphisms are related to alleviating cone death in RP [38], and to retinal degeneration in AMD [39]. Further studies are warranted to determine the possibility of *TGFB**I* as a causative gene for RP.

### 4.5. Genetic Testing and Inconclusive Cases

From our study, we identified seven cases with inconclusive results, and in most cases, we were unable to confirm appropriate genotype–phenotype correlations or incomplete penetrance. In a self-reporting asymptomatic family member, identification of a variant in the genes such as *GNAT1*, and *SNRNP200* with autosomal dominant inheritance, the possibility of non-penetrance or incomplete penetrance should be considered, and this asymptomatic family member should undergo thorough ophthalmologic examinations with long-term follow-up. Additionally, in cases of incompatible genotype–phenotype correlations, such as in those carrying *CRB1* or *GNAT1* variants without the typical RP phenotype, reassessment using WES is required, because there remains the possibility of unrevealed RP genes.

### 4.6. Limitations

This study has several limitations. First, we had to rely on patients’ reports for their clinical history, which could have led to recall bias. Second, most of patients with *USH2A* mutations in this study did not undergo the additional hearing tests. Even though most of the patients did not have subjective hearing loss, we have limited values in the clinical interpretation and classification of patients with *USH2A*. Third, segregation analysis was insufficient, especially to analyze inconclusive cases of variants with incomplete penetrance. It was possible to estimate the causative gene using pedigree analysis and clinical characteristics in most cases; however, we should emphasize the value of segregation analysis in de novo variants with asymptomatic family members or AD and XL-related diseases with carrier family members. In this context, we cannot rule out the possibility that VUSs, which did not undergo segregation analysis, are not the definite causative genes of the patients. Although we tried to compensate for this issue by examining the phenotypic consistency of RP and the correlation of posterior probabilities of the Bayesian calculator, further segregation analysis is needed. Lastly, the TGS panel must be partially reinforced for the differential diagnosis of IRDs in cases with overlapping phenotypes to reveal unexpected variants identified using WES. Considering that an increase in the number of genes constituting the gene panel does not necessarily increase the diagnostic rate, it is necessary to optimize the composition of the gene panel to maximize its clinical usefulness.

## 5. Conclusions

This study provides reliable and valuable information about the distribution of RP-related genes, and about the relationship between genes and the age of onset for RP. In addition, these results not only present genomic spectra for RP but also will serve as educational insights for clinicians regarding genetic testing in the differential diagnosis of IRDs and RP with overlapping or incompletely manifested phenotypes.

## Figures and Tables

**Figure 1 genes-12-00675-f001:**
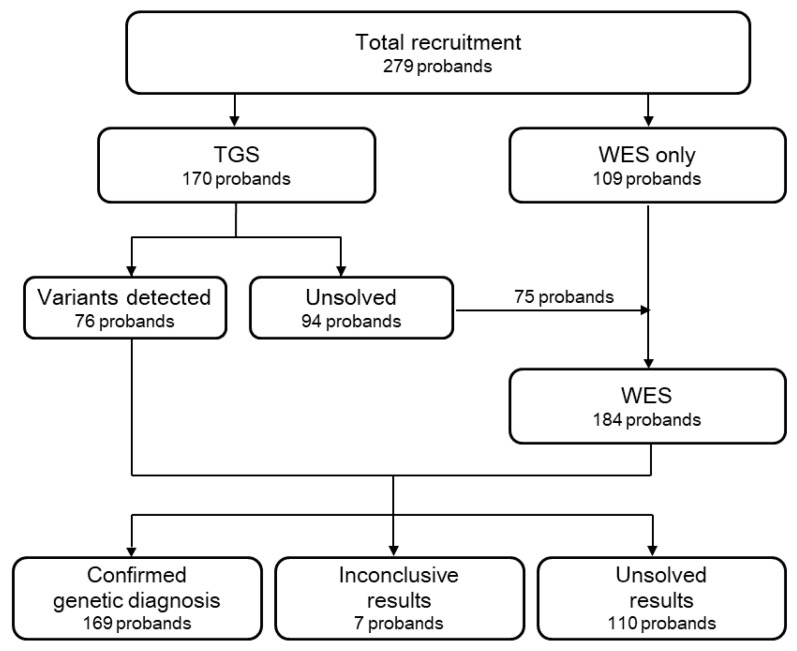
Flowchart demonstrating the recruitment progress of genetic tests in patients with retinitis pigmentosa.

**Figure 2 genes-12-00675-f002:**
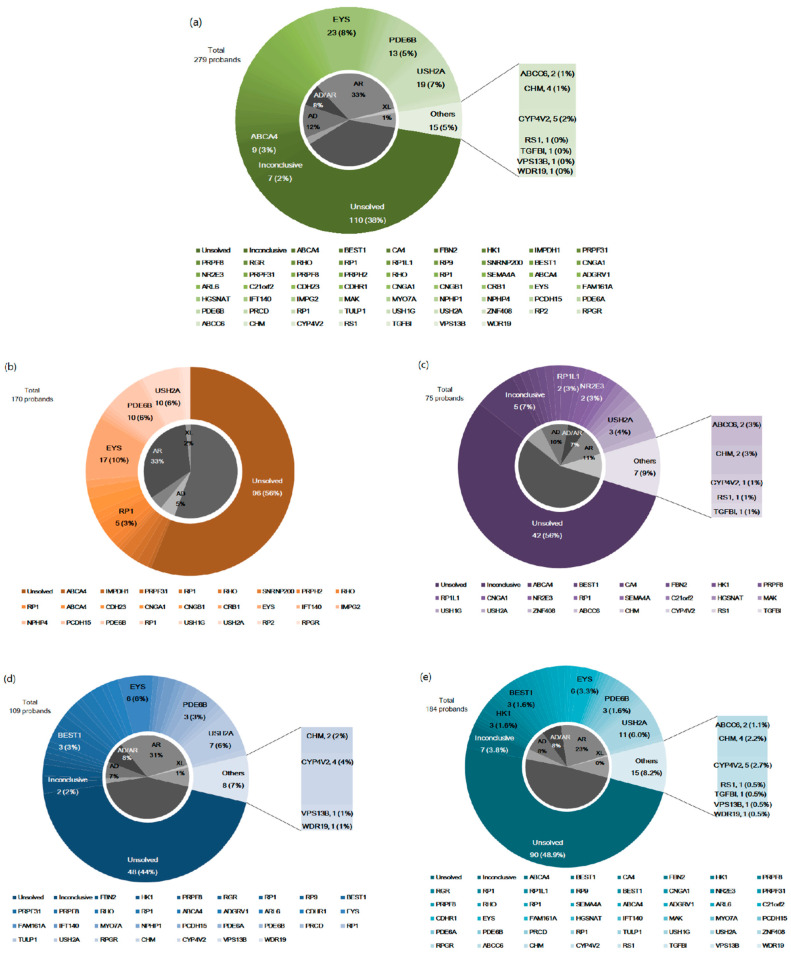
Mutational spectra of 279 probands with retinitis pigmentosa. (**a**) Mutational spectrum of total subjects, combining targeted next-generation sequencing (TGS) and whole exome sequencing (WES). (**b**) Mutational spectrum of patients who underwent TGS. (**c**) Mutational spectrum of patients who underwent WES due to TGS with inconclusive results. (**d**) Mutational spectrum of patients who underwent WES only. (**e**) Mutational spectrum of total patients who underwent WES with inconclusive TGS results and WES only. AD, autosomal dominant; AR, autosomal recessive; XL, x-linked inheritance.

**Figure 3 genes-12-00675-f003:**
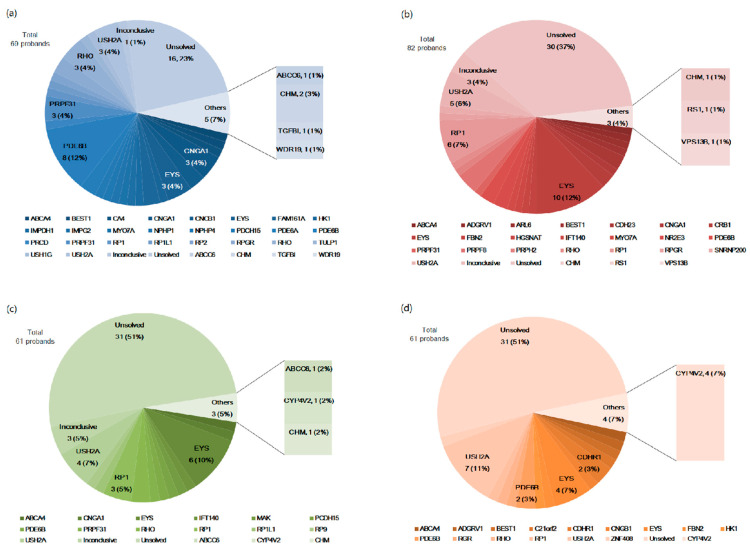
Mutational spectra of 279 probands with retinitis pigmentosa according to age at symptom onset. (**a**) Mutational spectrum for age at symptom onset under 10 years. (**b**) Mutational spectrum for age at symptom onset between 11 and 20 years. (**c**) Mutational spectrum for age at symptom onset between 21 and 40 years. (**d**) Mutational spectrum for age at symptom onset over 41 years.

**Table 1 genes-12-00675-t001:** Baseline clinical characteristics of probands with retinitis pigmentosa and subgroup analysis based on age at symptom onset.

	No. of Probands, n (%)	Sex, M:F (%)	Family History, n (Y:N, %)	Age at Genetic Examination, Years	Age at Symptom Onset, Years	Age at Diagnosis, Years	BCVA, LogMAR	
							RE	LE
Total subjects	279	131:148 (47.0:53.0)	92:187 (33.0:67.0)	47.6 ± 15.7	25.6 ± 16.9	41.1 ± 15.2	0.8 ± 1.0	0.8 ± 1.0
**Subgroup analysis according to age at symptom onset**								
≤10 years	69 (24.7)	41:28 (59.4:40.6)	25:44 (36.2:63.8)	40.6 ± 16.7	7.7 ± 2.1	30.7 ± 14.7	0.9 ± 1.1	0.8 ± 1.0
11–20 years	82 (29.4)	43:39 (52.4:47.6)	28:54 (34.1:65.9)	41.6 ± 14.8	16.0 ± 3.1	36.7 ± 13.8	0.9 ± 1.1	0.9 ± 1.1
21–40 years	61 (21.9)	21:40 (34.4:65.6)	19:42 (31.1:68.9)	52.2 ± 12.8	33.2 ± 5.5	44.6 ± 10.4	0.6 ± 0.9	0.7 ± 1.0
≥41 years	61 (21.9)	23:38 (37.7:62.3)	19:42 (31.1:68.9)	59.9 ± 7.9	51.0 ± 6.2	55.9 ± 7.9	0.7 ± 1.0	0.7 ± 0.9
asymptomatic	6 (2.1)	3:3 (50.0:50.0)	1:5 (16.7:83.3)	39.2 ± 15.4	N/A	36.2 ± 13.0	0.1 ± 0.1	0.0 ± 0.2

Abbreviations: M, male; F, female; SD, standard deviation; BCVA, best-corrected visual acuity; LogMAR, logarithm of the minimum angle of resolution; RE, right eye; LE, left eye; N/A, non-available.

**Table 2 genes-12-00675-t002:** Diagnostic yields of the genetic tests in patients with retinitis pigmentosa.

		No. of Probands, n	No. of Segregation Analyses, n	No. of Probands with Detected Variants, n (Detection Rate, %)	No. of Probands with Inconclusive Results, n (%)
Total subjects		279	60	161 (57.7)	10 (3.6)
Genetic analysis	TGS	170	29	74 (43.5)	
	WES from unsolved TGS	75	23	28 (37.3)	8 (10.7)
	WES only	109	31	59 (54.1)	2 (1.8)
Age at symptom onset	≤10 years	69	19	52 (75.4)	2 (2.9)
	11–20 years	82	19	49 (59.8)	4 (4.9)
	21–40 years	61	11	27 (44.3)	4 (6.6)
	≥41 years	61	10	30 (49.2)	0 (0.0)
	asymptomatic	6	1	3 (50.0)	0 (0.0)
Family history	+	91	28	65 (71.4)	2 (2.2)
	-	188	32	96 (51.1)	8 (4.3)

Abbreviations: TGS, targeted next-generation sequencing; WES, whole exome sequencing.

**Table 3 genes-12-00675-t003:** List of patients with unexpected genetic results causing inherited retinal diseases other than retinitis pigmentosa.

Subject No.	Causative Gene	NM Number	Chromosome	HGVS DNA	HGVS Protein Change	Zygosity	Inheritance	Origin	Chromosome	ACMG Criteria	
6-6	RS1	NM_000330.3	X	c.78+1G>A		Hemi	XL	Unknown	LPV	PVS1, PM2	Novel
7-7	CYP4V2	NM_207352.3	4	c.809_810C		Hetero	AR	Unknown	PV	PVS1, PM2, PP5	
	CYP4V2	NM_207352.3	4	c.992A>C	p.His331Pro	Hetero	AR	Unknown	LPV	PS1, PM2, PP2, PP3	
14-17	ABCC6	NM_001171.5	16	c.3703C>T	p.Arg1235Trp	Hetero	AD	Maternal	LPV	PS1, PM2, PP3	
21-28	TGFBI	NM_000358.2	5	c.371G>A	p.Arg124His	Hetero	AD	Unknown	PV	PS1, PS3, PM1, PM5, PP2, PP3	
21-29	TGFBI	NM_000358.2	5	c.371G>A	p.Arg124His	Hetero	AD	Unknown	PV	PS1, PS3, PM1, PM5, PP2, PP3	
22-30	CHM	NM_000390.3	X	c.688delinsTG		Hemi	XL	Maternal	LPV	PVS1, PM2	Novel
22-31	CHM	NM_000390.3	X	c.688delinsTG		Hemi	XL	Unknown	LPV	PVS1, PM2	Novel
61-96	ABCC6	NM_001171.5	16	c.3698T>C	p.Val1233Ala	Hetero	AD	Unknown	VUS	PM1, PM2, PP3	
72-112	CHM	NM_000390.3	X	c.1718_1719del	p.Tyr573CysfsTer12	Hemi	XL	Unknown	PV	PVS1, PM2, PP5	
72-113	CHM	NM_000390.3	X	c.1718_1719del	p.Tyr573CysfsTer12	Hemi	XL	Unknown	PV	PVS1, PM2, PP5	
172-223	WDR19	NM_025132.3	4	c.2645+1G>T		Hetero	AR	Unknown	LPV	PVS1, PM2	
	WDR19	NM_025132.3	4	c.1613G>T	p.Gly538Val	Hetero	AR	Unknown	VUS	PM2, PP3	Novel
188-252	CYP4V2	NM_207352.3	4	c.1072G>T	p.Glu358Ter	Homo	AR	Unknown	LPV	PVS1, PM2	
193-261	CHM	NM_000390.3	X	c.2T>A	p.Met1Lys	Hemi	XL	Unknown	VUS	PVS1_M, PM2	Novel
207-280	VPS13B	NM_017890.4	8	c.7220_7221A		Hetero	AR	Maternal	PV	PM2, PP3, BP1	
	VPS13B	NM_017890.4		c.11468G>C	p.Gly3823Arg	Hetero	AR	Unknown	VUS	PVS1, PM2, PP5	Novel
225-311	CYP4V2	NM_207352.3	4	c.809_810C		Homo	AR	Unknown	PV	PVS1, PM2, PP5	
240-337	CYP4V2	NM_207352.3	4	c.802_807A		Hetero	AR	Unknown	LPV	PVS1, PM2	
	CYP4V2	NM_207352.3		c.219T>A	p.Phe73Leu	Hetero	AR	Unknown	VUS	PM2, PP2, PP3	
248-349	CYP4V2	NM_207352.3	4	c.675-1G>A		Hetero	AR	Unknown	LPV	PVS1, PM2	Novel
	CYP4V2	NM_207352.3	4	c.802-8_807del		Hetero	AR	Unknown	LPV	PVS1, PM2	

Abbreviations: HGVS, Human Genome Variation Society; ACMG, American College of Medical Genetics and Genomics; AD, autosomal dominant; AR, autosomal recessive; XL, x-linked inheritance; PV, pathogenic variant; LPV, likely pathogenic variant; VUS, variant of unknown significance.

## Data Availability

The datasets generated during and/or analyzed during the current study are available from the corresponding author on reasonable request.

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
