# Peer review of "Diverse Genetic Landscape of Suspected Retinitis Pigmentosa in a Large Korean Cohort"

_genes, 2021, doi:10.3390/genes12050675_

Round 1

Reviewer 1 Report

The manuscript by Kim et al. describes the mutational landscape of retinitis pigmentosa in South Korea, by a combined approach of whole-exome and panel-NGS sequencing in 279 unrelated patients who were clinically diagnosed with this condition.

The manuscript presents novel experimental data, is well written, and is very easy to follow. The conclusions are perfectly justified by the results obtained.

We only have minor comments.

1) New data have been recently published on the prevalence of RP and other IRDs, based on population-wide genetic information (Hanany et al., PNAS 2020). The authors may therefore want to update the information provided in the first paragraph of the Introduction.

2) FSCN2 has been recognized not to be an IRD-associated gene since a long time. The text should be amended to remove all information relative to this gene.

3) It would be helpful to add an extra column in all Tables listing the variants identified, reporting whether or not such variants have been reported in ClinVar as being clearly pathogenic.

Author Response

Reviewer 1

The manuscript by Kim et al. describes the mutational landscape of retinitis pigmentosa in South Korea, by a combined approach of whole-exome and panel-NGS sequencing in 279 unrelated patients who were clinically diagnosed with this condition.

The manuscript presents novel experimental data, is well written, and is very easy to follow. The conclusions are perfectly justified by the results obtained.

We only have minor comments.

New data have been recently published on the prevalence of RP and other IRDs, based on population-wide genetic information (Hanany et al., PNAS 2020). The authors may therefore want to update the information provided in the first paragraph of the Introduction.

  • Thank you for your comments. We updated information regarding the IRD causing genes to the Introduction section of the revised manuscript as follows: Inherited retinal diseases (IRDs) are a group of heterogenous conditions in which progres-sive visual impairment is caused by retinal degeneration, and are mainly caused by Men-delian mutations in 1 out of at least 300 genes[1,2]. (line 37-39)

FSCN2 has been recognized not to be an IRD-associated gene since a long time. The text should be amended to remove all information relative to this gene.

  • As per your recommendation, we removed all information related to FSCN2 (Table 3 and Table S5)

It would be helpful to add an extra column in all Tables listing the variants identified, reporting whether or not such variants have been reported in ClinVar as being clearly pathogenic.

  • We added a column which demonstrates the ACMG evidence codes and novel mutations to Table 3 and Table S4, S5.

Reviewer 2 Report

This work describes next generation sequencing of 279 Korean individuals with RP (and IRD). A strategy with initial screening with a targeted panel and subsequent exom sequencing (WES). Some individuals only underwent WES. This work is of great importance especially because of the ethnic group that is investigated.

The manuscript needs further work before publication. I will make some general point that authors need to improve (and not make specific comments before the next version). But the manuscript need a thoroughly work through.

Some genes are wrongly written, i.e. TGFB1 where the authors probably mean TGFBI, which corresponds to NM_000358.2 and which is associated with corneal dystrophy. A gene called CNB1 does not exists, and the transcript number NM_000945 listed to be associated with the gene is a transcript for PPP3R1. Authors need to the carefully check all genes and NM numbers. I would suggest that authors make a table with gene names, NG numbers and NM numbers and avoid having the NM numbers in the text. This will make the text more readable.

In the discussion authors should be careful not to repeat the results. The discussion could improve by a major revision.

Comments to the tables:

Inheritance should indicate the known inheritance for the genes. That is, if only one variant is found in ABCA4 the inheritance should NOT be listed as AD but should be AR since it is known that ABCA4 variants are inherited in a autosomal recessive manner. All genes in the table should follow this.

The criteria for the ACMG classification should be listed in the table. For example “PM2” if the variant are not found in controls, to make it transparent how the classification was made.

Comments to figures:

Figure 1: The box with total recruitment should be on the top of the figure, and in the bottom of the figure should be the number of individuals with a confirmed genetic diagnosis, and partial diagnosis.

Figure 2 and 3: The text should be with a larger font size. I would suggest that authors are more restrictive in selection of which data to show.

Author Response

Comments and Suggestions for Authors

This work describes next generation sequencing of 279 Korean individuals with RP (and IRD). A strategy with initial screening with a targeted panel and subsequent exom sequencing (WES). Some individuals only underwent WES. This work is of great importance especially because of the ethnic group that is investigated.

The manuscript needs further work before publication. I will make some general point that authors need to improve (and not make specific comments before the next version). But the manuscript need a thoroughly work through.

Some genes are wrongly written, i.e. TGFB1 where the authors probably mean TGFBI, which corresponds to NM_000358.2 and which is associated with corneal dystrophy. A gene called CNB1 does not exists, and the transcript number NM_000945 listed to be associated with the gene is a transcript for PPP3R1. Authors need to the carefully check all genes and NM numbers. I would suggest that authors make a table with gene names, NG numbers and NM numbers and avoid having the NM numbers in the text. This will make the text more readable.

  • Thank you for your meticulous comments. As you pointed out, the gene ‘TGFB1’ has been modified to be ‘TGFBI’. In addition, ‘CNB1’ and its corresponding NM numbers were typographical errors and modified to ‘CNGA1’ and ‘NM_000087’.

As the reviewer suggested, we deleted the NM numbers in the manuscript and made a list of NM numbers and the genes mentioned in the analysis as Table S4 for the readability of the manuscript.

In the discussion authors should be careful not to repeat the results. The discussion could improve by a major revision.

  • We revised the Discussion section to concisely provide the interpretation and clinical implications of our results as the reviewer suggested.

Comments to the tables:

Inheritance should indicate the known inheritance for the genes. That is, if only one variant is found in ABCA4 the inheritance should NOT be listed as AD but should be AR since it is known that ABCA4 variants are inherited in a autosomal recessive manner. All genes in the table should follow this.

  • We fully agree with the reviewer’s comment. Since the phenotypes of RP and the AD inherited ABCA4 are not clinically inconsistent, we reclassified the patients with AD inherited ABCA4 mutation from the 'inconclusive results' group to the 'unsolved' group. Therefore, we deleted genetic results of ABCA4 and their clinical features from Table S5 and Figure S2 in the revised manuscript.

The criteria for the ACMG classification should be listed in the table. For example “PM2” if the variant are not found in controls, to make it transparent how the classification was made.

  • We added a column which demonstrates the ACMG evidence codes to Table 3 and Table S4, S5.

Comments to figures:

Figure 1: The box with total recruitment should be on the top of the figure, and in the bottom of the figure should be the number of individuals with a confirmed genetic diagnosis, and partial diagnosis.

  • Following your recommendation, we modified the arrangement of the flow of the Figure 1.

Figure 2 and 3: The text should be with a larger font size. I would suggest that authors are more restrictive in selection of which data to show.

  • We changed the font size to be larger and changed the position of legends in the figures to improve visibility (Figure 2, 3).

Reviewer 3 Report

The authors reported abundant and relevant genomic information on 279 retinal dystrophy probands. However, the authors discussed mutations in IRD genes and inconclusive cases other than mutations in RP genes.  There’s some unclear information that I have listed below. I recommend accepting this case study with subject to major revisions. Following are my comments:

  1. Please provide Sanger sequencing results for the reported IRD gene mutations in supplementary file.
  2. Please add the missing figure 2e legend. 
  3. Figures should be modified; I recommend authors to increase pixels. All mutation reported genes and their respective prevalence percentage findings must be within the figure. In the center of each figure (figure 2 and 3), authors should give information on the total number of mutations reported by WES/TGS or by both which indicate that out of how many mutations, authors calculated these percentages.
  4. Authors should also add a paragraph discussing how many mutations are previously reported and how many are novel in their findings.
  5. I highly recommend authors move the genetic information table (Table S4) in the main manuscript as it contains useful information. 
  6. In line 252, please mention the reference for Chinese and Taiwanese population. 
  7. As authors discussed information on fundus and OCT pictures, phenotypes for patients carrying IRD mutations. I highly recommend authors add ERG reports of cases also as it will give more clarity on the phenotypes.
  8. Please discuss about the diverse functions of RD genes in visual pathway, metabolism, signaling in the introduction. Use reference https://doi.org/10.1007/s10633-018-9654-x. 
  9. Authors mentioned the prevalence of USH2A gene mutations in their study but did not discuss the USH2A gene in the introduction. I strongly recommend authors use reference from DOI:10.1016/j.jcjo.2018.02.008 to discuss RP categorization, syndromic type of RP in the introduction section. 
  10. USH2A is also linked with hearing disorder, authors should also mention associated anomaly reported in their cases. Do their cases belong to type 1, type 2 or type 3 USH category in discussion section.

Author Response

Comments and Suggestions for Authors

The authors reported abundant and relevant genomic information on 279 retinal dystrophy probands. However, the authors discussed mutations in IRD genes and inconclusive cases other than mutations in RP genes.  There’s some unclear information that I have listed below. I recommend accepting this case study with subject to major revisions. Following are my comments:

Please provide Sanger sequencing results for the reported IRD gene mutations in supplementary file.

  • We appreciate your thoughtful comments. As we mentioned in the Methods section in the original manuscript, we validated the genetic test results through Sanger sequencing in all patients who underwent WES whereas validation with Sanger sequencing was waived in the TGS test based on the read depth of the test. Thus, all results recorded in Table 3, S4, and Table S5 were those validated by Sanger sequencing.

Please add the missing figure 2e legend.

  • We added the missing legend of Figure 2e (e) Mutational spectrum of total patients who underwent WES with inconclusive TGS results and WES only) to the revised manuscript.

Figures should be modified; I recommend authors to increase pixels. All mutation reported genes and their respective prevalence percentage findings must be within the figure. In the center of each figure (figure 2 and 3), authors should give information on the total number of mutations reported by WES/TGS or by both which indicate that out of how many mutations, authors calculated these percentages.

  • We changed the font size to be larger and changed the position of legends in the figures to improve visibility (Figure 2, 3). In addition, we added the total number of variants reported by WES/TGS or both to the center of each figure and each number of patients confirmed a genetic diagnosis along with their percentages.

Authors should also add a paragraph discussing how many mutations are previously reported and how many are novel in their findings.

  • As per your recommendation, we added the information regarding the novel variants in this study as follows: Among 181 detected variants, 43 variants were identified as novel variants (line 136-137).

I highly recommend authors move the genetic information table (Table S4) in the main manuscript as it contains useful information.

  • As your recommendation, we moved Table S4 to Table 3 in the revised main manuscript to provide the useful information to the readers.

In line 252, please mention the reference for Chinese and Taiwanese population.

  • To clearly demonstrate the reference for Chinese and Taiwanese population, we changed the location of reference number and clarify the proportion of the variants in Chines and Taiwanese populations those are similar to Western and European ethnicities as follows: On the other hand, USH2A variants make up 20–40% and EYS variants account for less than 10% of RP cases,[6,18], among the total RP population in Western or European ethnicities[8,19,20] as well as Chinese or Taiwanese populations[7, 21] (line 259-261).

As authors discussed information on fundus and OCT pictures, phenotypes for patients carrying IRD mutations. I highly recommend authors add ERG reports of cases also as it will give more clarity on the phenotypes.

  • We added the ERG findings to Table S1 and S2 in the revised manuscript.

Please discuss about the diverse functions of RD genes in visual pathway, metabolism, signaling in the introduction. Use reference https://doi.org/10.1007/s10633-018-9654-x.

  • We added sentences about the various functions of IRD genes in relation to clinical features or ethnic diversity to the Introduction section in the revised manuscript as follows: More than 80 genes have been identified as responsible for RP [1,4]. Diverse functions of RP causative genes involve various pathways, i.e. phototransduction, vitamin A metabo-lism, signalling, cell–cell interaction, and protein synthesis, i.e. structural or cytoskeletal proteins, synaptic interaction proteins, mRNA intron-splicing factors, trafficking of intra-cellular proteins, maintenance of cilia/ciliated cells, phagocytosis, pH regulator and a few encode proteins with yet unknown function[5] (line 42-47).

Authors mentioned the prevalence of USH2A gene mutations in their study but did not discuss the USH2A gene in the introduction. I strongly recommend authors use reference from DOI:10.1016/j.jcjo.2018.02.008 to discuss RP categorization, syndromic type of RP in the introduction section.

  • We added sentences explaining the syndromic RP in the Introduction section of revised article as follows: In addition, genetic abnormalities expressed in various organs other than eyes, cause syndromic RP such as Usher syndrome [6] (line 47-49).

USH2A is also linked with hearing disorder, authors should also mention associated anomaly reported in their cases. Do their cases belong to type 1, type 2 or type 3 USH category in discussion section.

  • We totally agree with your pointing out. It is important to evaluation hearing problems in patients with the USH2A Unfortunately, most of patients who participated in this study, however, did not consent to the additional hearing tests. Therefore we could not accurately diagnose Usher syndrome. We mentioned this limitation in the Discussion section of revised manuscript as follows: 4.6 Limitations. Second, most of patients with USH2A mutations in this study did not underwent the additional hearing tests. Even though most of patients did not have subjective hearing loss, we have limited values in clinical interpretation and classification of patients with USH2A (line 353-356).

Round 2

Reviewer 3 Report

The authors have addressed all my comments effectively. The manuscript now looks good for publication.